# Early Markers of Cardiovascular Disease Associated with Clinical Data and Autosomal Ancestry in Patients with Type 1 Diabetes: A Cross-Sectional Study in an Admixed Brazilian Population

**DOI:** 10.3390/genes13020389

**Published:** 2022-02-21

**Authors:** Roberta Maria Duailibe Ferreira Reis, Rossana Santiago de Sousa Azulay, Maria da Glória Tavares, Gilvan Cortês Nascimento, Sabrina da Silva Pereira Damianse, Viviane Chaves de Carvalho Rocha, Ana Gregória Almeida, Débora Cristina Ferreira Lago, Vandilson Rodrigues, Marcelo Magalhães, Carla Souza Sobral, Conceição Parente, Joana França, Jacqueline Ribeiro, Paulo Cézar Dias Ferraz, Carlos Alberto Azulay Junior, Dayse Aparecida Silva, Marília Brito Gomes, Manuel dos Santos Faria

**Affiliations:** 1Graduate Program in Adult Health (PPGSAD), Federal University of Maranhão-UFMA, Av. dos Portugueses, São Luís 65085-580, Brazil; magalhaes_ms@yahoo.com (M.M.); mfaria1949@gmail.com (M.d.S.F.); 2Service of Endocrinology, University Hospital of the Federal University of Maranhão (HUUFMA/EBSERH), Rua Barão de Itapary, 227-Centro, São Luís 65020-070, Brazil; rossanaendocrino@gmail.com (R.S.d.S.A.); madagloria@gmail.com (M.d.G.T.); gilvancortes@uol.com.br (G.C.N.); sabrinadamianse@gmail.com (S.d.S.P.D.); vcc2007@terra.com.br (V.C.d.C.R.); debora.lago@ufma.br (D.C.F.L.); carlasouzasobral@gmail.com (C.S.S.); conceicao.parente@huufma.br (C.P.); joana-franca@hotmail.com (J.F.); 3Research Group in Clinical and Molecular Endocrinology and Metabology (ENDOCLIM), São Luís 65020-070, Brazil; agfpalmeida@gmail.com (A.G.A.); vandilson.rodrigues@ufma.br (V.R.); jacqueribeiro.cardio@uol.com.br (J.R.); pc1987@gmail.com (P.C.D.F.); carlosazulayjr@gmail.com (C.A.A.J.); 4DNA Diagnostic Laboratory (LDD), Rio de Janeiro State University (UERJ), R. São Francisco Xavier, Rio de Janeiro 20550-013, Brazil; dayse.a.silva@gmail.com; 5Diabetes Unit, State University of Rio de Janeiro (UERJ), R. São Francisco Xavier, Rio de Janeiro 20550-013, Brazil; mariliabgomes@gmail.com

**Keywords:** type 1 diabetes, cardiovascular disease, calcium score, carotid Doppler, ankle-brachial index, ancestry

## Abstract

Patients with type 1 diabetes (T1D) have a higher risk of developing cardiovascular disease (CVD), which is a major cause of death in this population. This study investigates early markers of CVD associated with clinical data and autosomal ancestry in T1D patients from an admixed Brazilian population. A cross-sectional study was conducted with 99 T1D patients. The mean age of the study sample was 27.6 years and the mean duration of T1D was 14.4 years. The frequencies of abnormalities of the early markers of CVD were 19.6% in the ankle-brachial index (ABI), 4.1% in the coronary artery calcium score (CACS), and 5% in the carotid Doppler. A significant percentage of agreement was observed for the comparison of the frequency of abnormalities between CACS and carotid Doppler (92.2%, *p* = 0.041). There was no significant association between the level of autosomal ancestry proportions and early markers of CVD. The ABI was useful in the early identification of CVD in asymptomatic young patients with T1D and with a short duration of disease. Although CACS and carotid Doppler are non-invasive tests, carotid Doppler is more cost-effective, and both have limitations in screening for CVD in young patients with a short duration of T1D. We did not find a statistically significant relationship between autosomal ancestry proportions and early CVD markers in an admixed Brazilian population.

## 1. Introduction

The incidence of cardiovascular disease (CVD) is two to four times higher in patients with type 1 diabetes (T1D) than in the general population [1,2]. It has been demonstrated that the atherosclerotic process starts early, and that its late clinical manifestations can be atypical or silent [3,4]. Some studies have reported that CVDs become the leading cause of death after about 20 years of duration of type 1 diabetes, with a significant association between diabetes duration and CVD (independent of patient age) and the threshold effect perceived at approximately 20 years [5]. Advances have been made in the management of microvascular complications in patients with T1D, but there has been little progress in reducing CVD [4]. Therefore, the early identification of CVD could reduce the morbidity and mortality of T1D patients.

Subclinical CVD has usually been evaluated through non-invasive methods, such as computed tomography with the coronary artery calcium score (CACS), carotid Doppler ultrasound, and ankle-brachial index (ABI). Changes in these tests have been associated with an increase in CVD and may be useful in the early identification of this pathology [6,7].

Identifying clinical factors associated with these early markers of CVD could help physicians better monitor T1D patients. Furthermore, the influence of ancestry on the development and progression of CVD in T1D remains unclear. A study comparing South Asian and European people with T1D showed higher mortality in people from South Asia, inferring the possible role of ethnicity in the development of CVD and mortality [8]. However, these studies did not use autosomal ancestry, which may result in some biases. In the assessment of autosomal ancestry, informative ancestry markers (AIMs) can be used, which are especially useful in determining individual ancestry and degrees of miscegenation. The analysis of these polymorphisms are determined through the selection of autosomal markers such as SNPs and also insertions/deletions (INDELs) [9,10]. As far as we know, there are no data in mixed populations, such as the Brazilian population, examining the relationship between CVD, T1D, and proportions of autosomal ancestry.

The aim of this study was to investigate early markers of CVD associated with clinical data and autosomal ancestry in T1D patients from an admixed Brazilian population.

## 2. Materials and Methods

### 2.1. Study Design and Sample

This was a cross-sectional study conducted with T1D patients from the University Hospital of the Federal University of Maranhão (HU-UFMA), São Luís, Brazil. The study conformed to the ethical guidelines of the 1975 Declaration of Helsinki, and the project was approved by the Research Ethics Committee of the HU-UFMA (no. 2.668.396). All participants or their legal representatives were informed about the study objectives and procedures and signed an informed consent form.

The study sample included T1D patients aged above 10 years recruited from August 2018 to October 2019 at the Diabetes Clinic of the HU-UFMA. The diagnosis of T1D was defined by clinical history, based polyuria, polydipsia, polyphagia, and weight loss associated with insulin use since diagnosis. Non-inclusion criteria were as follows: pregnancy, lactation, previous acute myocardial infarction and stroke, history of myocardial revascularization, angioplasty, and known peripheral arterial disease (PAD). A total of 99 patients met the eligibility criteria and were accepted to participate in the study.

### 2.2. Anthropometric and Laboratory Data

T1D patients were given a clinical-demographic survey through a standardized questionnaire, in which data were collected on sex, age (years), age at T1D diagnosis (years), and duration of T1D duration (years). The following variables were analyzed: weight (in kilograms), height (in centimeters), body mass index (BMI), systemic blood pressure (BP), and waist circumference (determined at half the distance between the last costal arch and the iliac crest). The following laboratory variables were determined: fasting blood glucose (enzymatic), glycated hemoglobin (high performance liquid chromatography), hs-CRP (immunoturbidimetry), urea (colorimetric), creatinine (colorimetric), total cholesterol (enzymatic), HDL (enzymatic) calculated LDL by the Friedewald equation [11], and the urinary albumin concentration (immunoturbidimetry) through the collection of a random urine sample on up to three occasions, considering the cutoff value of 30 mg/dL. The variables were analyzed as continuous variables.

### 2.3. Ophthalmological Evaluation

Retinal evaluation was performed through examination of the fundus of the eye by two different examiners, and classified as normal, non-proliferative retinopathy (mild, moderate and severe), proliferative retinopathy, and diabetic maculopathy [12].

### 2.4. Analysis of Early CVD Markers

The ABI was measured by a single examiner using a sphygmomanometer and stethoscope. The BP of the right and left upper limb was determined by palpating the brachial artery, and in the right and left lower limb by palpation of the posterior tibialis. Subsequently, the ratio between the lower limbs over the upper was calculated, and a value between 0.9 and 1.3 was considered normal [13].

Computed tomography for evaluation of CACS was performed using the Aquilion TSX-101A 64-channel device (Toshiba Medical Systems, Tokyo, Japan), obtained in non-contrast-enhanced acquisition of a series of 3 mm thick axial slices covering the entire length of the heart. It was ranked in the following ranges 0, 0–10, 10–100, 100–400, >400 Agatston [14]. Values above 0 were considered abnormal.

Carotid Doppler ultrasound was accomplished by the same examiner using a Logic E ultrasound device (GE Healthcare, Wisconsin, USA), with a 12 cm (12L-RS) linear transducer and a frequency of 5.0–13.0 MHz, using the central frequency of 7.5 MHz Three measurements were averaged on each side of the common carotids. Patients who presented at least one site larger than 1.5 mm were classified according to the plate. Patients with a mean between 0.9 and 1.5 mm were classified as having thickening [15]. The presence of thickening or plaque was considered abnormal.

### 2.5. Autosomal Ancestry Proportions

To infer the European, African, and Amerindian ancestry proportions, a panel of 46 autosomal informational insertion/deletion ancestry markers (AIM–Indels), amplified in a single multiplex polymerase chain reaction (PCR), was used according to the protocol described by Pereira et al. [16]. The identification of polymorphisms in the generated fragments was achieved by capillary electrophoresis in the ABI 3500 automatic sequencer (Life Technologies). Genotyping was performed by two independent analysts using GeneMapper Analysis Software v. 4.1 (Life Technologies), and the outcomes were compared for consistency. Structure v. 2.3.3 software was utilized to estimate ancestry and the Human Genome Diversity Genotype Database–Centre d’Étude du Polymorphisme Humain (HGDP–CEPH) panel was used as a reference for ancestral populations [17]. The allele frequencies of 46 genotyped AIM–INDELs were compared with a database of a healthy Brazilian population for the same markers from all geographic regions of Brazil [17].

### 2.6. Statistical Analysis

Data were analyzed using SPSS version 28.0 (IBM, Chicago, IL, USA) and GraphPad Prism version 9.1.1 (GraphPad Software, Inc., San Diego, CA, USA). Descriptive statistics included measures of absolute frequency, percentage, mean, and standard deviation (±sd). The overall percentage of agreement was calculated to compare the different methods for detecting cardiovascular abnormalities, and the probability value was estimated by Fleiss’ Kappa statistics. Categorical variables were analyzed using the chi-square test and Fisher’s exact test. The Shapiro–Wilk and D’Agostino–Pearson tests were used to assess the normality of the distribution. Comparative analysis of continuous variables were performed with an independent *t*-test. The significance level was set at 0.05.

## 3. Results

A total of 99 patients with T1D (56 women), with a mean age of 27.6 ± 10.2 years, were included in the present study. In the sample, the mean age at diagnosis of T1D was 14.4 ± 8.4 years, with mean T1D duration of 13.2 ± 8.3 years. The prevalence of microalbuminuria and retinopathy was 17.3% and 27.3%, respectively. Regarding the evaluation of autosomal ancestry proportions, European ancestry was the main component in our sample (47.3 ± 14.1), followed by African (28 ± 12.6) and Amerindian (24.7 ± 9.4). Clinical data and laboratory evaluations are summarized in Table 1.

The frequencies of abnormalities of the early CVD markers were 4.1% in the CACS, 5% in the Doppler, and 19.2% in the ABI (Table 2). A significant percentage of agreement was observed between CACS and Doppler (92.2%, *p* = 0.041), and between Doppler and ABI (79.7%, *p* = 0.005) (Figure 1).

Abnormal CACS was statistically higher among patients with microalbuminuria ≥ 30 mg/dL (17.7% versus 1.3%, *p* = 0.017). In addition, Doppler abnormality was found to be higher among patients with retinopathy (15.4% versus 1.5%, *p* = 0.020) (Table 3).

Table 4 summarizes the result of the association analysis between the continuous variables evaluated in the study and cardiovascular outcomes. Older patients (*p* = 0.001), with longer duration of T1D (*p* = 0.002) and with higher levels of LDL cholesterol (*p* = 0.041), were more likely to be found in the CACS abnormality group (Figure 2). Older patients (*p* < 0.001), with higher systolic blood pressure (*p* = 0.019) and longer T1D duration (*p* < 0.001), were often observed in the group with Doppler alteration (Figure 3). Older patients (*p* = 0.041) with greater waist circumference (*p* = 0.026) were found in patients with altered ABI (Figure 4).

In the sample of T1D patients evaluated, there was no significant association between the level of autosomal ancestry proportions and early markers of cardiovascular disease (Table 5 and Figure 5).

In addition, there was no significant correlation between clinical data, such as HbA1c, LDL, hs-CRP, triglycerides, DPB, FBG, and autosomal ancestry proportions (Figure 6).

## 4. Discussion

Our study demonstrated that the prevalence of early CVD lesions varied according to the method used, which is useful for assessing atherosclerotic disease in several phases. ABI, carotid Doppler, and CACS are methods of cardiovascular risk (CVR) stratification. Among these, ABI showed the highest prevalence of alterations in our series, making it useful in the early identification of CVD in young asymptomatic patients with T1D and with a short period of disease. We observed an agreement between the detection of cardiovascular injury using the ABI and carotid Doppler methods and between CACS and carotid Doppler, the latter with a low prevalence of alterations in our sample. The study finding suggests that there is no association between ancestry and early CVD markers.

The literature shows a higher prevalence of CVD in women with T1D, in contrast with the general population, where the greatest risk is in males [4]. In this study, the prevalence of alterations between genders varied according to the method used, with a higher prevalence of females found with ABI and carotid Doppler, although the differences were not statistically significant.

Tests such as ABI, carotid Doppler, and CACS are useful in the stratification of cardiovascular risk, but they are not routinely indicated in the international guidelines of the American Diabetes Association and European Association for the Study of Diabetes for screening in asymptomatic patients in the stratification of CVR in patients with diabetes [18,19]. However, the Brazilian guidelines for prevention of cardiovascular disease in patients with diabetes recommends the use of these screening methods in the asymptomatic population [5]. The MESA study compared the performance of these different stratification methods in an intermediate-risk population without previous cardiovascular events (Framingham score between 5% and 20%), and CACS was superior to Doppler for predicting the risk of coronary events, and both were superior to the ABI. However, this study was not focused on patients with diabetes [20]. In our assessment, we found a prevalence of ABI alteration of 19.2%, CACS of 4.1%, and carotid Doppler of 5%, reinforcing the importance of these methods in the stratification of CVD in T1D.

Studies in diabetics show the importance of performing ABI in asymptomatic patients in order to detect PAD [21,22]. Studies performing ABI in asymptomatic T1D patients found alterations between and 32% and 33% of their samples, reinforcing the importance of detecting subclinical PAD [22,23]. Our prevalence was 19.2%, and showed agreement with carotid Doppler, an expected finding due to both being useful for the evaluation of endothelial dysfunction. In the present study, abnormal ABI was associated with higher age and higher abdominal circumference. A cohort study with 5.003 older adults reinforced the finding of ABI < 1.2 and higher risk of CVD, reinforcing its importance in primary prevention [24]. ABI is an easy-to-perform and low-cost test, which can be useful in earlier stages of CVD detection and as a screening tool for CVR stratification when there are still no changes in other tests also used for this purpose.

Meta-analysis studies in patients with T1D show a thickening of the carotid intima-media thickness (CMIT) by carotid Doppler sonography, even in non-diagnostic values of subclinical disease from very young ages [25,26]. Factors that can accelerate the specific CMIT increase process are controversial in the literature; however, in a recent meta-analysis, worse glycemic control was associated with higher CMIT [25]. In our analysis, the patients who presented abnormalities in the test were outside the recommended target for glycemic control (HbA1c > 7%), but this value was not statistically significant. We observed an association between CMIT thickening and increased waist circumference, which may reflect the role of visceral adipose tissue in endothelial dysfunction and CVR, as also suggested by other authors [27].

The evaluation of CACS in T1D has been more studied in recent years, as it is a non-invasive test with great sensitivity to indirectly measure the global coronary atherosclerotic load, and it has excellent outcomes when combined with the Framingham parameters [28]. Aguilera et al. evaluated CACS and carotid Doppler in T1D and, despite the high degree of agreement between them, concluded that due to the low prevalence of lesions and its high cost, CACS should not be used for screening for CVD in patients with T1D lasting less than 20 years [29]. In our sample, 4.1% had CACS > 0 and 5% had carotid Doppler abnormalities. This low prevalence can be explained by the young age (average of 28 years old) and short time of disease (average of 14 years) of our patients. A large cohort study with a median follow-up time of 4.3 years and the median age of 58 (49–65) years found that CACS 0 excluded CVD dependent on age, beyond clinical variables, and the added diagnostic value was lower for younger patients, corroborating the importance of age for better accuracy when estimating the risk of CVD [30]. However, despite the low prevalence, there was a significant association between abnormal CACS and thickening or plaque on doppler examination with older age, reflecting the importance of this risk factor for the development of CVD [31]. There was an association of CACS with albuminuria and of Doppler with retinopathy, which was also seen in other studies [30,31], reaffirming the association between micro and macrovascular complications in patients with diabetes.

Studies have demonstrated the importance of glycemic control and reduced risk of microvascular complications [32,33]. However, in relation to macrovascular disease, the Diabetes Control and Complications Trial (DCCT) showed a non-significant reduction in patients in the intensive glycemic control group. Although, in the follow-up of these patients in the Epidemiology of Diabetes Interventions and Complications (EDIC), there was a substantial reduction in non-fatal events in this intensive control group, reinforcing the importance of the time factor for the reduction of macrovascular disease [32,33]. In our study, no association was found between early CVD markers and HbA1c, which may be due to the short duration of the disease and young age in our sample. In addition, there was no association with other biochemical markers such as HDL, triglycerides, hs-CRP, urea, and creatinine.

Ancestry data in patients with T1D showed a predominance of incidence in white people of European origin (47%) [7,8]. A study carried out in the state of Maranhão also showed mainly European origin and suggested that the increased risk of T1D may have been transmitted in the process of miscegenation by European ancestors. We found agreement with the studies mentioned above, showing a higher prevalence of European origin in our patients with T1D [34]. Studies in type 2 diabetes have shown the influence of ethnicity on the development of CVD, and when comparing different populations, an impact of ethnicity on the development of CVD has been suggested, but these studies did not use autosomal ancestry proportions, which can lead to bias [35,36]. In our study, we did not find an association between autosomal ancestry and CVD markers. Further studies are required to assess the relationship between ancestry and CVD risk in T1D patients, with larger sample size and a longitudinal design, and to better evaluate the nature of these associations over time and life stages. Analysis of paternal (Y chromosome) and maternal (mitochondrial DNA) ancestry patterns may be useful to assess whether these genetic markers are related to CVD risk.

Other studies show that Afro-descendants have lower visceral fat, higher HDL, and lower triglycerides when compared to white, and in contrast, the former group has a higher rate of hypertension and insulin resistance [35,36]. However, when evaluating clinical data such as metabolic syndrome, glycemic control, and renal disease in patients with T1D in Brazil, no significance was found between these clinical data and ancestry data as in our analysis [37,38,39,40].

We consider the sample size and population assessed (young and with a short duration of T1D) as limitations in our study, as the incidence of atherosclerotic disease is usually low in this group. However, this study evaluated different methods together for the detection of CVD in patients with T1D to provide differentiation of these methods, aiming at the development of better procedures for the early detection of CVD, and a consequent reduction of morbidity and mortality in this population. We also added the unprecedented evaluation of these early CVD markers with genetic ancestry in a mixed population.

## 5. Conclusions

In conclusion, the findings suggest that the ABI can be useful in the early identification of CVD in asymptomatic young patients with T1D and with a short disease duration, and it is among the most cost-effective tests for the early markers of cardiovascular disease. Examinations, such as carotid Doppler and CACS, showed significant agreement between them and were correlated with known risk factors for CVD in patients with T1D. Although both tests are non-invasive, carotid Doppler is more cost-effective, and both have limitations in screening for CVD in patients with a short duration of T1D and young age. We found no association between autosomal ancestry and early CVD markers in the study sample. Further studies are needed to identify when to use early CVD stratification methods in asymptomatic patients with T1D, and to further explore the influence of autosomal ancestry on the development of CVD.

## Figures and Tables

**Figure 1 genes-13-00389-f001:**
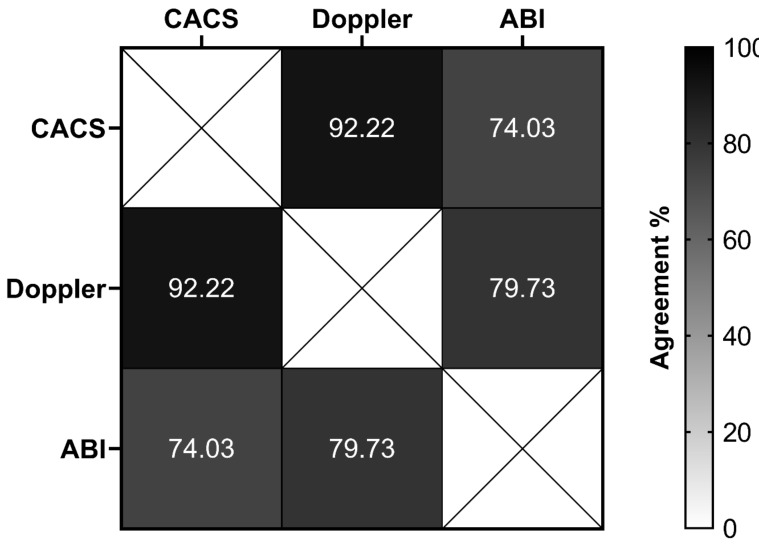
The overall percentage of agreement between different methods to evaluate early markers of cardiovascular disease in type 1 diabetes patients. CACS, coronary artery calcium score (normal versus abnormal); Doppler, carotid Doppler sonography (normal versus abnormal); ABI, ankle–brachial index (normal versus abnormal).

**Figure 2 genes-13-00389-f002:**
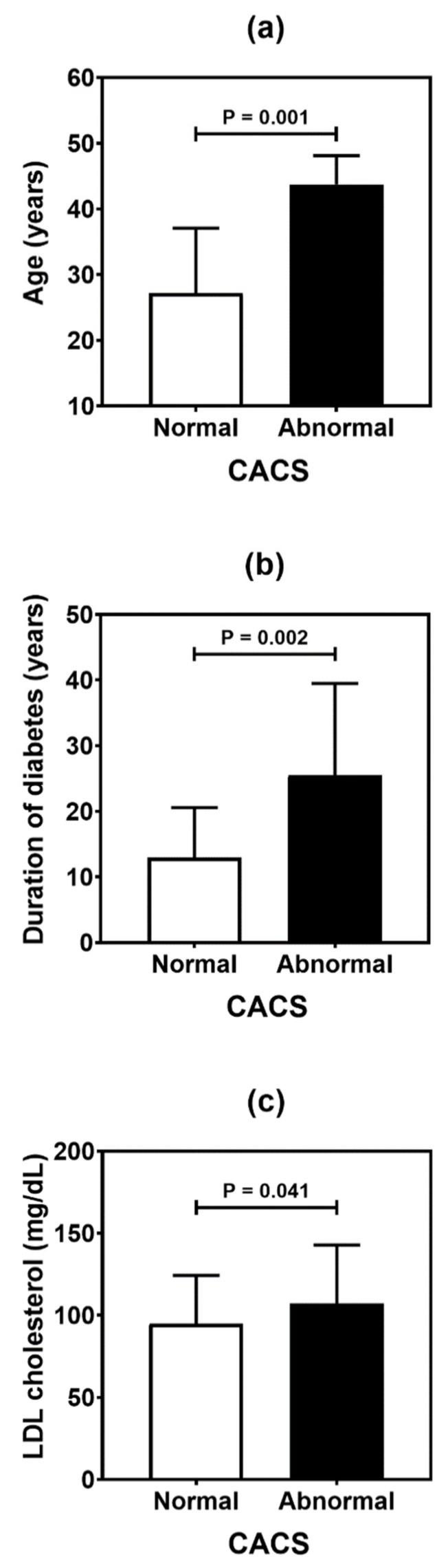
Comparative analysis of age (**a**), duration of diabetes (**b**), and LDL (low-density lipoprotein) cholesterol level (**c**) in type 1 diabetes patients according to the coronary artery calcium score (CACS). *p* values were determined by independent *t*-tests.

**Figure 3 genes-13-00389-f003:**
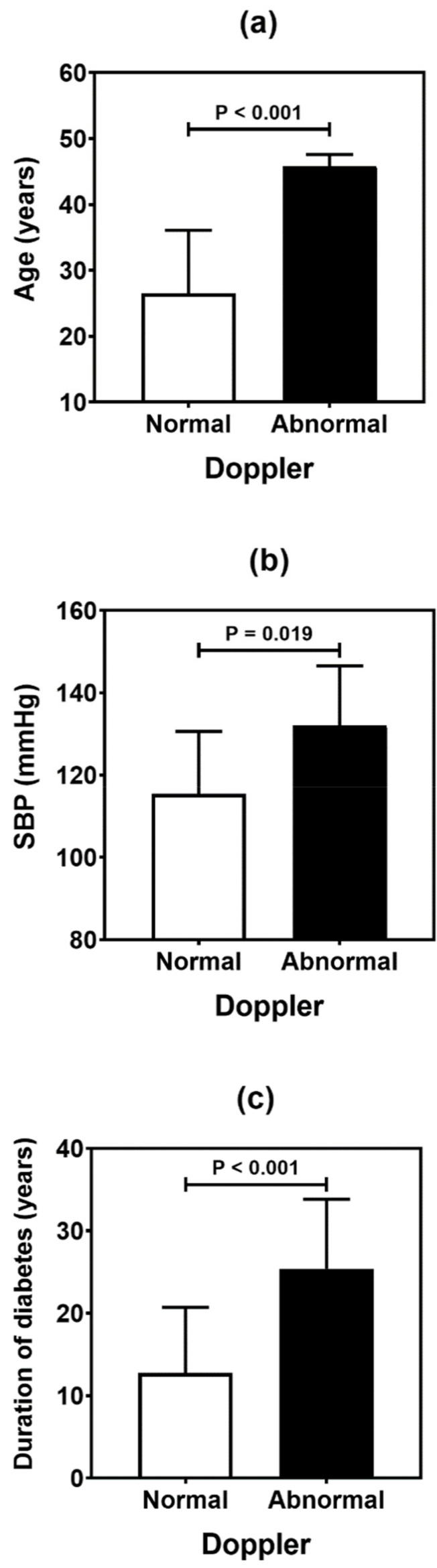
Comparative analysis (mean and standard deviation) of age (**a**), systolic blood pressure (SBP) (**b**), and duration of diabetes (**c**) in type 1 diabetes patients according to the carotid Doppler sonography. *p* values were determined by independent *t*-tests.

**Figure 4 genes-13-00389-f004:**
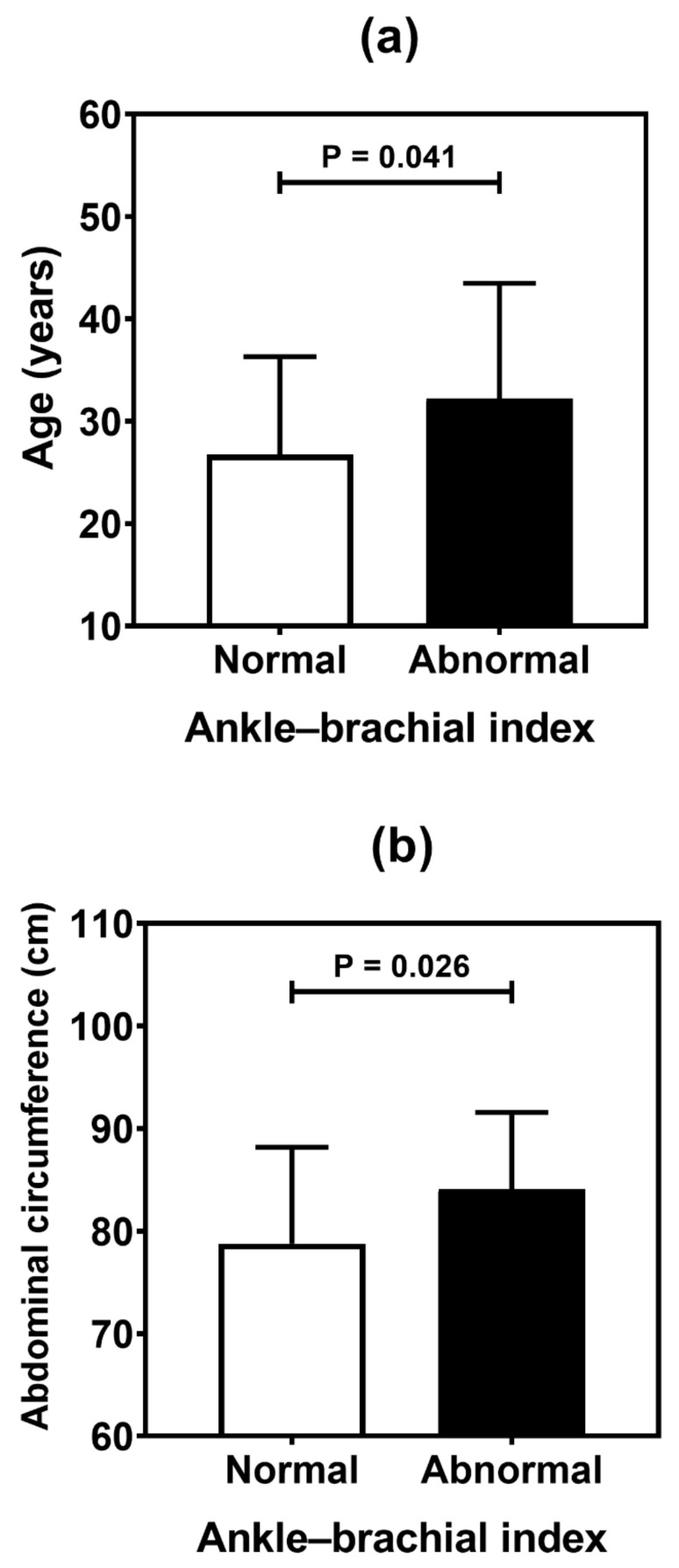
Comparative analysis (mean and standard deviation) of age (**a**), and abdominal circumference (**b**) in type 1 diabetes patients according to the ankle-brachial index. *p* values were determined by independent *t*-tests.

**Figure 5 genes-13-00389-f005:**
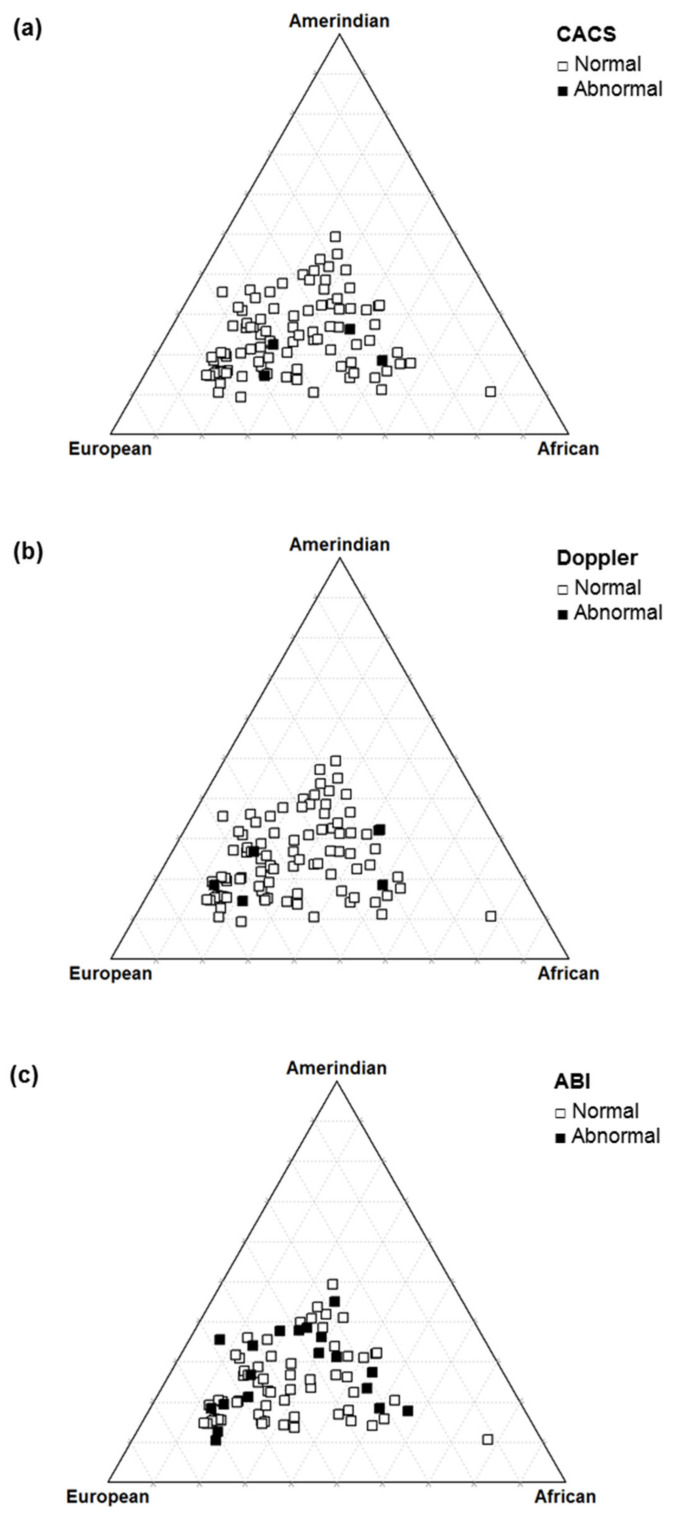
Triangle plot of autosomal ancestry in type 1 diabetes patients according to the coronary artery calcium score (**a**), carotid Doppler sonography (**b**), and ankle–brachial index (**c**).

**Figure 6 genes-13-00389-f006:**
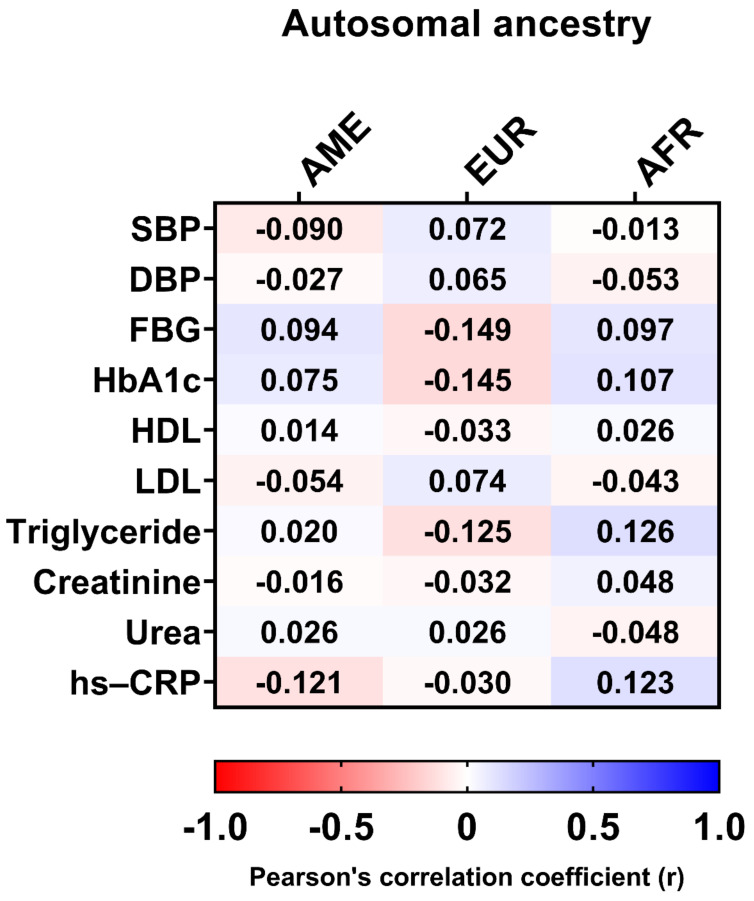
Pearson’s correlation coefficients between clinical data and autosomal ancestry. AME, Amerindian; EUR, European; AFR, African; SBP, systolic blood pressure; DBP, diastolic blood pressure; FBG, fasting blood glucose; HbA1c, glycated hemoglobin; HDL, high-density lipoprotein; LDL, low-density lipoprotein; hs-CRP, high-sensitivity C-reactive protein.

**Table 1 genes-13-00389-t001:** Demographics and clinical data of type 1 diabetes patients.

Variables	*n* (%)	Mean ± sd
Age (in years)		27.6 ± 10.2
Sex		
Female	56 (55.6)	
Male	43 (43.4)	
Anthropometric data		
Body mass index (kg/m^2^)		22.5 ± 3.8
Abdominal circumference in female group (cm)		82.1 ± 13.8
Abdominal circumference in male group (cm)		89.1 ± 8.5
Autosomal ancestry proportions		
Amerindian		24.7 ± 9.4
European		47.3 ± 14.1
African		28.0 ± 12.6
Arterial pressure		
Systolic blood pressure (mmHg)		116.4 ± 15.5
Diastolic blood pressure (mmHg)		74.1 ± 10.4
Diabetes data		
Age at T1D Diagnosis (years)		14.4 ± 8.4
Duration of diabetes (years)		13.2 ± 8.3
Fasting glucose (mg/dL)		188.3 ± 108.1
HbA1c (%)		8.9 ± 2.2
Microalbuminuria		
<30 mg/dL	81 (81.8)	
≥30 mg/dL	17 (17.2)	
No data	1 (1.0)	
Retinopathy		
Absent	72 (72.7)	
Present	27 (27.3)	
Creatinine clearance		
≥60 mL/min	95 (95.9)	
<60 mL/min	4 (4.1)	
Serum data		
LDL cholesterol (mg/dL)		96.9 ± 31.7
HDL cholesterol (mg/dL)		54.0 ± 13.5
Triglyceride (mg/dL)		104.6 ± 65.7
Creatinine (mg/dL)		0.86 ± 0.64
Urea (mg/dL)		28.8 ± 13.2
hs-CRP (mg/L)		0.25 ± 0.38

±sd, standard deviation; T1D, type 1 diabetes; HbA1c, glycated hemoglobin; HDL, high–density lipoprotein; LDL, low–density lipoprotein; hs-CRP, high–sensitivity C–reactive protein.

**Table 2 genes-13-00389-t002:** Distribution of early markers of cardiovascular disease in the type 1 diabetes patients.

Variables	*n*	(%)
Coronary artery calcium score		
Normal	92	(92.9)
Abnormal	4	(4.1)
No data	3	(3.0)
Carotid Doppler sonography		
Normal	88	(88.9)
Abnormal	5	(5.0)
No data	6	(6.1)
Ankle–brachial index		
Normal	60	(60.6)
Abnormal	19	(19.2)
No data	20	(20.2)

**Table 3 genes-13-00389-t003:** Comparative analysis of cardiovascular abnormalities detected by different methods in type 1 diabetes patients according to sex, microalbuminuria, and retinopathy.

Variables	Cardiovascular Abnormalities Detected by					
	CACS	*p* Value	Doppler	*p* Value	ABI	*p* Value
	%		%		%	
Gender (Sex)		1.000		0.373		0.116
Female	3.8		7.8		30.4	
Male	4.7		2.4		15.2	
Microalbuminuria		0.017 *		0.540		1.000
<30 mg/dL	1.3		3.9		23.4	
≥30 mg/dL	17.7		6.3		21.4	
Retinopathy		1.000		0.020 *		0.786
Absent	4.3		1.5		23.2	
Present	3.7		15.4		26.1	

CACS, coronary artery calcium score; Doppler, carotid Doppler sonography; ABI, ankle–brachial index. * Significant differences (*p* < 0.05).

**Table 4 genes-13-00389-t004:** Results of comparison tests (*p* values) of the clinical and serum data in type 1 diabetes patients according to early markers of cardiovascular disease.

Variables	Cardiovascular Abnormalities		
	CACS	Doppler	ABI
	*p* Value	*p* Value	*p* Value
Age	0.001 *	<0.001 *	0.041 *
Anthropometric data			
Body mass index	0.268	0.154	0.518
Abdominal circumference	0.256	0.052	0.026 *
Arterial pressure			
Systolic blood pressure	0.160	0.019 *	0.570
Diastolic blood pressure	0.351	0.164	0.629
Diabetes data			
Age at T1D diagnosis	0.299	0.111	0.145
Duration of diabetes	0.002 *	<0.001 *	0.172
Fasting glucose	0.278	0.916	0.571
HbA1c (%)	0.637	0.608	0.716
Serum data			
LDL cholesterol	0.041 *	0.222	0.639
HDL cholesterol	0.948	0.557	0.138
Triglyceride	0.750	0.979	0.976
Creatinine	0.873	0.553	0.395
Urea	0.729	0.361	0.989
hs-CRP	0.516	0.648	0.426

T1D, type 1 diabetes; HbA1c, glycated hemoglobin; HDL, high-density lipoprotein; LDL, low-density lipoprotein; hs-CRP, high-sensitivity C–reactive protein; CACS, coronary artery calcium score (normal versus abnormal); Doppler, carotid Doppler sonography (normal versus abnormal); ABI, ankle–brachial index (normal versus abnormal). * Significant differences (*p* < 0.05) by independent *t*-test or Mann-Whitney test.

**Table 5 genes-13-00389-t005:** Results of comparison tests (*p* values) of autosomal ancestry data in individuals with type 1 diabetes according to a cardiovascular evaluation.

Variables	Cardiovascular Abnormalities		
	CACS	Doppler	ABI
	*p* Value	*p* Value	*p* Value
Autosomal ancestry			
Amerindian	0.222	0.795	0.765
European	0.383	0.481	0.234
African	0.688	0.801	0.636

CACS, coronary artery calcium score (normal versus abnormal). Doppler, carotid doppler sonography (normal versus abnormal). ABI, ankle–brachial index (normal versus abnormal).

## Data Availability

The datasets used and analyzed during the current study are available from the corresponding author upon reasonable request.

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
