# Peer review of "Early Markers of Cardiovascular Disease Associated with Clinical Data and Autosomal Ancestry in Patients with Type 1 Diabetes: A Cross-Sectional Study in an Admixed Brazilian Population"

_genes, 2022, doi:10.3390/genes13020389_

Round 1

Reviewer 1 Report

This is an interesting study of early markers of CVD associated with clinical data and autosomal ancestry in T1D patients. I would like to ask you to reconsider some things to make a better paper.

Minor 1; In page 11, it is shown in Table S1, but isn't it Table 5?

Major 1; In page 14, it says "more studies are needed to assess the relationship between ancestry and the risk of CVD in T1D.", but I would like to know what kind of research you are planning.

Major 2; When considering the involvement of autosomal ancestry, I think that good or bad glycemic control can be a bias in these T1D patients. Is it possible to exclude this from the results of this research?

Author Response

Dear Editor,

We appreciate all the reviewer’ comments. We have modified the manuscript in accordance with these recommendations, “Early markers of cardiovascular disease associated with clinical data and autosomal ancestry in patients with type 1 diabetes: A cross-sectional study in an admixed Brazilian population” (Manuscript ID: genes-1574850).

Reply to reviewers:

Reviewer #1: Minor 1; In page 11, it is shown in Table S1, but isn't it Table 5?

Response: We have corrected the text accordingly.

Reviewer #1: Major 1; In page 14, it says "more studies are needed to assess the relationship between ancestry and the risk of CVD in T1D.", but I would like to know what kind of research you are planning.

Response: We thank the reviewer’s comments for improving our manuscript. We have modified the sentence to clarify this point.

“More studies are needed to assess the relationship between ancestry and CVD risk in DM1. Analyzes of paternal (Y chromosome) and maternal (mitochondrial DNA) ancestry patterns may be useful to assess whether these genetic markers are related to CVD risk.”  

Reviewer #1: Major 2; When considering the involvement of autosomal ancestry, I think that good or bad glycemic control can be a bias in these T1D patients. Is it possible to exclude this from the results of this research?

Response: We have understood the reviewer's concern. However, we have included the figure 6 to evaluate correlation between autosomal ancestry and clinical data. There was no significant correlation between HbA1c and autosomal ancestry.

Yours sincerely,

Roberta Reis

Reviewer 2 Report

This article investigated the association of CVD markers with clinical data and autosomal ancestry in T1D patients, which is meaningful and interesting. However, I think some data should be added.

1) The specific value of CACS, carotid thickness measure by Carotid Doppler ultrasound and ABI should be given.

2) The relationship of the specific values of CVD markers with clinical data and autosomal ancestry should be further investigated. 

Author Response

February 03, 2022

Editor-in-Chief

Dr. Mohsen Ghanbari,

Genes

Dear Editor,

We appreciate all the reviewer’ comments. We have modified the manuscript in accordance with these recommendations, “Early markers of cardiovascular disease associated with clinical data and autosomal ancestry in patients with type 1 diabetes: A cross-sectional study in an admixed Brazilian population” (Manuscript ID: genes-1574850).

Reply to reviewers:

Reviewer #2: The specific value of CACS, carotid thickness measure by Carotid Doppler ultrasound and ABI should be given.

Response: Thank you for your comments. The values considered in each of the methods were placed in the item "Methods", in the sub-item "Analysis of early CVD markers".

Reviewer #2: The relationship of the specific values of CVD markers with clinical data and autosomal ancestry should be further investigated.

Response: We agree with the reviewer’s recommendation. We have added the figure 6 in the revised manuscript in order to evaluate the correlation between autosomal ancestry and clinical data and added a discussion in our text.

“Other studies show that Afro-descendants have lower visceral fat, higher HDL and lower triglycerides when compared to white, and in contrast, the former group has a higher rate of hypertension and insulin resistance [35,36]. However, when evaluated clinical data such as metabolic syndrome, glycemic control and renal disease in patients with T1D in Brazil, no significance was found between these clinical data and ancestry data as in our analysis [37,38,39].”

Yours sincerely,

Roberta Reis

Reviewer 3 Report

Review of the article “ Early markers of cardiovascular disease associated with clinical data and autosomal ancestry in patients with type 1 diabetes: A cross-sectional study in an admixed Brazilian population.”

In the present study the authors have studied the relation between ankle-brachial index (ABI), coronary artery calcium score (CACS) and carotid Doppler in subjects with Type 1 diabetes with short duration of the disease. The authors have also studied the possible relation between those three tests to autosomal ancestry. The authors found that Abi was useful in the early detection of cardiovascular disease (CVD) and that there was no relationship between autosomal ancestry proportions and early CVD.

The manuscript is well written and high-lights the important aspect of early detection of CVD in young patients with Type 1 diabetes.

Issue #1: It is interesting that there was no relation of HbA1c, one of the most used markers in clinical practise, to early CVD. There was no relation between CVD and other biochemical markers such as HDL-cholesterol, triglycerides, creatine, urea or hs-CRP to only older patients with CVD abnormalities had higher levels of LDL cholesterol.

I think it worth to mention in the discussion that these biochemical markers were not very useful as indicators of cardiovascular abnormalities in these young patients with Type 1 diabetes and short duration of the disease.

Author Response

February 03, 2022

Editor-in-Chief

Dr. Mohsen Ghanbari,

Genes

Dear Editor,

We appreciate all the reviewer’ comments. We have modified the manuscript in accordance with these recommendations, “Early markers of cardiovascular disease associated with clinical data and autosomal ancestry in patients with type 1 diabetes: A cross-sectional study in an admixed Brazilian population” (Manuscript ID: genes-1574850).

Reply to reviewers:

Reviewer #3: It is interesting that there was no relation of HbA1c, one of the most used markers in clinical practice, to early CVD. There was no relation between CVD and other biochemical markers such as HDL-cholesterol, triglycerides, creatinine, urea or hs-CRP to only older patients with CVD abnormalities had higher levels of LDL cholesterol. I think it worth to mention in the discussion that these biochemical markers were not very useful as indicators of cardiovascular abnormalities in these young patients with Type 1 diabetes and short duration of the disease.

Response: Thank you for your comments. We found it pertinent and added a further discussion of biochemical markers in our text.  

“Studies have demonstrated the importance of glycemic control and reduced risk of microvascular complications [30,31]. However, in relation to macrovascular disease, the DCCT (Diabetes Control and Complications Trial) showed a non-significant reduction in patients in the intensive glycemic control group. Although, in the follow-up of these patients in the EDIC (Epidemiology of Diabetes Interventions and Complications), there was a substantial reduction in non-fatal events in this intensive control group, reinforcing the importance of the time factor for the reduction of macrovascular disease [30,31]. In our study, no association was found between early CVD markers and Hb1ac, which may be due to the short duration of the disease and young age in our sample. In addition, there was no association with other biochemical markers such as HDL, triglycerides, hs-CRP, urea, and creatinine.”

Yours sincerely,

Roberta Reis

Round 2

Reviewer 2 Report

The authors have addressed my concerns properly.

Author Response

February 13th, 2022

The entire manuscript was carefully revised by a native English speaker and typos were corrected.

The review certificate is indexed.

Yours sincerely,

Roberta Reis
